# Mitigating the Impact of the COVID-19 Pandemic on Adult Cancer Patients through Telehealth Adoption: A Systematic Review

**DOI:** 10.3390/s22093598

**Published:** 2022-05-09

**Authors:** Aileen Murphy, Ann Kirby, Amy Lawlor, Frances J. Drummond, Ciara Heavin

**Affiliations:** 1Department of Economics, Cork University Business School, University College Cork, T12 CY82 Cork, Ireland; aileen.murphy@ucc.ie (A.M.); a.kirby@ucc.ie (A.K.); alawlor@ucc.ie (A.L.); 2Breakthrough Cancer Research, T12 F9XD Cork, Ireland; frances@breakcancer.ie; 3Department of Business Information Systems, Cork University Business School, University College Cork, T12 CY82 Cork, Ireland

**Keywords:** cancer, oncology, telehealth, COVID-19, pandemic, patients, systematic review

## Abstract

During the first wave of the COVID-19 pandemic, the delivery of life-saving and life-prolonging health services for oncology care and supporting services was delayed and, in some cases, completely halted, as national health services globally shifted their attention and resources towards the pandemic response. Prior to March 2020, telehealth was starting to change access to health services. However, the onset of the global pandemic may mark a tipping point for telehealth adoption in healthcare delivery. We conducted a systematic review of literature published between January 2020 and March 2021 examining the impact of the COVID-19 pandemic on adult cancer patients. The review’s inclusion criteria focused on the economic, social, health, and psychological implications of COVID-19 on cancer patients and the availability of telehealth services emerged as a key theme. The studies reviewed revealed that the introduction of new telehealth services or the expansion of existing telehealth occurred to support and enable the continuity of oncology and related services during this extraordinary period. Our analysis points to several strengths and weaknesses associated with telehealth adoption and use amongst this cohort. Evidence indicates that while telehealth is not a panacea, it can offer a “bolstering” solution during a time of disruption to patients’ access to essential cancer diagnostic, treatment, and aftercare services. The innovative use of telehealth has created opportunities to reimagine the delivery of healthcare services beyond COVID-19.

## 1. Introduction

Health systems worldwide pivoted towards telehealth when the COVID-19 pandemic started [1]. This means that a range of consultations and services was delivered via telephone, video conferencing, and other messaging services [2], extending from consultations with general practitioners in primary care to specialist consultants in hospitals across medical areas, including cancer care [3]. This was particularly pertinent for cancer patients and survivors who are at greater risk of infection and developing more severe health complications compared to the general population, due to their weakened immune systems [3]. Globally, telehealth combined with other public health guidelines and initiatives, such as dedicated pathways and hubs, helped protect cancer patients and survivors from exposure to COVID-19.

In the current literature, a range of terms is used to describe the use of information and communication technology (ICT) to either deliver health services, transfer information, and/or provide education to patients and healthcare professionals at a distance [4]. For this study, we use the term “Telehealth” to characterise two-way communications among stakeholders [5] and their use of ICT [6,7] to enable the delivery of healthcare, exchange of health data, and dissemination of health-related education across geographically dispersed locations [8,9,10,11,12,13]. Research indicates that telehealth primarily focuses on the delivery of healthcare services [7,13,14], including the diagnosis, treatment, and prevention of disease and injuries [8]. The terms telehealth, telemedicine, and telecare are often used interchangeably. Telehealth is an overarching term involving clinical and non-clinical applications and encompasses both telecare and telemedicine [9]. Telecare occurs when a health-related request for assistance is made; a disease is not necessary to prompt such a request, and the other actor is not necessarily a healthcare professional [9]. Whereas telemedicine is typically more clinical, focusing on patient assessment, diagnosis, and treatment [9]. Colucci et al. [9] classify both telehealth and telemedicine as communication strategies whereby an action is taken remotely for the purpose of providing care or a cure. Telehealth also encompasses tele-radiology, tele-stroke, tele-ICU, tele-psychiatry, tele-burn, tele-prescription, and virtual care [15]. The role of healthcare professionals is core to the delivery of telehealth services [8].

Data from Google Trends (an online tool for comparison of the popularity of terms searched by users of Google Search and their trends over time) (Figure 1) illustrate the relative number of searches (based on the total volume of searches over time by geographic region) for the terms “Telehealth” and “Telemedicine” since 2017. As evidenced, these terms were widely searched for during March 2020, when the World Health Organization (WHO) declared COVID-19 a global pandemic, with Google Trends highlighting telehealth as the more popular search term.

The pattern illustrated in Figure 1 supports Edwards et al.’s [6] assertion that healthcare professionals are more familiar with the term telehealth compared with telemedicine. Indeed, March 2020 marks a time when frontline healthcare professionals scrambled to identify new ways to communicate with and deliver healthcare services to the most vulnerable patients [16].

There are many benefits associated with telehealth adoption; these include reported high levels of patient satisfaction with the telehealth experience [17], efficiency, privacy, comfort [18], and improved convenience, particularly for patients with physical limitations [19] as they can access important services from their own home. Other studies identify reduced fuel and parking costs and the cost of work absence [20] as financial benefits for patients. While, in their study of telehealth neonatal services, Ballantyne et al. [21] highlight avoiding traffic, not being late for appointments, and not having to arrange childcare as motivators for telehealth adoption. Advantages for healthcare providers include being able to access patients from their home offices [22] and reducing patient no-shows [23]. However, some studies identify challenges such as equipment costs, patient privacy, insufficient training, and limited communication between clinicians and information technology specialists [24].

We conducted a systematic literature review examining the social, psychological, health, and economic impacts of COVID-19 on cancer patients. This review captured the first wave of the pandemic, exploring the immediate responses of stakeholders to the crisis, thus revealing how healthcare services were taking steps to mitigate the effects of the pandemic for the most vulnerable patients. Telehealth emerged as a key theme; findings are presented here. This article is structured as follows; the next section describes the research materials and methods used and the systematic literature review execution. The following section reports the results and discussion. Finally, the conclusion considers limitations, implications arising from this study, and opportunities for future research.

## 2. Materials and Methods

The methodology for this systematic review was guided by Preferred Reporting Items for Systematic Reviews and Meta-Analyses (PRISMA) guidelines [25] and the population, intervention, comparators, outcomes, context, studies (PICOCS) framework was employed [26] (Table 1). The review’s inclusion criteria were limited to studies that focus on the economic, social, health, and psychological implications of COVID-19 on cancer patients/survivors; studies written in English and published between January 2020 and March 2021. The following types of studies were excluded: letters to the editor, editorials, case studies, reports, protocols, commentaries, short communications, reviews, opinions, perspectives, and discussions (PROSPERO registration number: CRD42021246651). The search strategy was developed using a combination of free text words and subject headings relevant to CINAHL, MEDLINE, PsycINFO, PsycArticles, and EMBASE platforms and was refined using Boolean operators. The search was conducted on 31 March 2021.

Data extraction is presented in a tabular format to assist in reporting uniformity and reproducibility and minimise bias. Included elements: general information: title, author(s), year of publication, country; study characteristics: aim/objective, perspective, study design, reason for inclusion; population characteristics: sample size, type of cancer, age, gender, patient, or survivor group; methods: context/setting, data source, study timeframe, data collection methods, data analysis methods; and outcome variables: economic impact, social impact, health-related impact, psychosocial impact. The JBI critical appraisal tools for cross-section, prevalence, qualitative, and cohort [27] as appropriate and CHEC list [28] for cost analyses were used to assess quality. The evidence was pooled and summarised to create an inventory of results with a narrative synthesis. Two authors (AL and AK) independently performed quality assessment. If there was conflict or uncertainty, a third author (AM) was consulted. Risk of bias in a study was considered high if the “yes” score was ≤4; moderate if 5–6; and low risk if the score was ≥7 on the JBI tools. Quality review results are presented in Appendix A. 

## 3. Results

### 3.1. Overview of Search Results

The search initially yielded 5383 studies, of which 167 were considered for full text review (see Murphy et al. [29] for further information). The review’s inclusion criteria focused on the economic, social, health, and psychological implications of COVID-19 on cancer patients and the availability of telehealth services emerged as a key theme (See Appendix B). Overall, 37 of the 167 articles reported on the use of telehealth in delivering oncology care during the COVID-19 pandemic (Figure 2). In this collection, many papers detailed experiences of telehealth in the USA (35%) and referred to services available for multiple/all cancer types (67%). Most of which were in a hospital setting (92%), two were in the community (5%), and one was web-based (3%). 

Telehealth services were employed in various stages of cancer treatment, including screenings/referrals [30]; radiation oncology [31,32,33,34,35,36,37]; surgical oncology [38,39,40,41]; follow-ups and counselling services [42,43,44,45,46,47,48]; rehabilitation services [49]; and palliative care [50,51,52]. Other studies considered telehealth across the delivery of oncology services in specific institutions [15,53,54,55,56,57,58]. Several studies assessed satisfaction/acceptability of telehealth services introduced standalone or as part of organisational changes during the pandemic [43,50,51,52,53,55,57,59,60]. One study considered the costs of delivering and accessing telehealth [61]. A range of study designs was reported, including retrospective, prospective, cross-sectional, and observational and a mix of primary and secondary data sources was employed (see Table 2 for study details).

### 3.2. Range of Telehealth Services Employed

In some instances, hospitals and health centres used and/or expanded existing telehealth services during the pandemic [15,32,40,50,51,53,54]. However, many studies demonstrated the development of new telehealth services in response to the pandemic. These telehealth initiatives facilitated the delivery of existing services [31,37,42,45,46,48,49,55,57,58] or development of new services [43,44,52]. The types of telehealth employed varied from video calls [52] or phone consultations [37,38,42,43,45,46,48,53,54,55] to a combination of phone, video, and texts [15,31,49,51,57,58].

Telehealth was employed as part of oncology services redesigned to ensure continuity of care and reduce risk of COVID-19 transmission [33,34,36,37,41,47,56]. Included in this were efforts to evaluate patients’ side effects/symptoms, avoid unnecessary hospital visits post-surgery [39], and reduce outpatient department workloads during the pandemic [38]. Where available, telehealth facilitated consultations for a variety of activities including, for patients seeking information regarding their oncological treatments, symptom management and replenishing opioid medications [51] or post-surgery to evaluate patients’ side effects [39]. In one instance, telehealth was employed to deliver tobacco treatment for tobacco-dependent cancer patients during the pandemic in New York City, which resulted in higher attendance and increased completion odds compared to in-person visits [42].

### 3.3. Satisfaction with Telehealth

The review demonstrates the rapid uptake of telehealth overall, and many studies found a high satisfaction rate amongst patients, caregivers, and clinicians [32,43,48,50,51,52,53,55,59]. Some patients expressed the desire to continue telehealth visits in the future [37,50,54,55]. Several studies outlined the advantages of using telehealth, which included increased access for patients and their families with regards to attendance at appointments [42,49,52,54] and decreased waiting lists [49]. Several authors suggested telemedicine was feasible given the circumstances of the pandemic, as it minimises disruptions to care and reduces patient risk of COVID-19 exposure [42,44,46,49,51,58], including avoiding visits to emergency departments [54] and saved time and costs from the hospital’s perspective [37,61]. Moreover, from a patient’s perspective, telehealth reduced travel time and expenses and increased convenience [45,54,55,58,61].

However, some patients preferred in-person visits and had no desire to continue telemedicine in the future, especially for staging results or treatment decisions [55,57]. Some studies highlight concerns about the accuracy and adequacy of virtual examinations (Canada [49] and Italy [60]) and their inconsistent use [31]. Others highlight that telehealth was not a substitute for treatment; for example, the use of telehealth did not mitigate the reductions in diagnosis and surgery (USA [65]). From the patients’ perspective, concerns about access barriers were also discussed. These included technical difficulties with internet access [42,46,53,54,64], costs of technologies/hardware [63], and communication barriers [49]. Several studies reported that telemedicine was less feasible for older patients as they tended to encounter more difficulties with technology [31,42,53,54,55]. Moreover, for specific activities such as genetic testing, requiring at-home sampling non-adherence was high [45]. Patient access barriers associated with telehealth were reported in developing economies such as network issues in India [64] and Brazil [46], and costs of technologies in Brazil [63]. Nevertheless, some institutions in developed countries also had access and adoption issues (USA, [65]) and concerns regarding the financial sustainability of providing telehealth (USA, [31]).

The review suggests some cancer patients and caregivers were very satisfied with telehealth and were comfortable using it as it offered them “support and connectedness” (India, [50], helped cope with worries [49], and reduced distress [53]. However, these favourable experiences were not universal and for some, the switch to telehealth was associated with higher levels of cancer worry and feelings of isolation (USA, [62]; Canada [49]). While some patients indicated a desire to continue with telehealth [37,55] and even expressed their willingness to pay for telehealth service in the future (India [50]), others indicated they were not inclined to continue using it. This was the case especially for staging results and treatment decisions [57], or it was only seen as a temporary measure until post-lockdown when the clinics reopened [35]. However, even those who were interested in continuing telehealth indicated a blend of telehealth and in-person visits is most desirable and suggested avoiding telehealth when providing results or prognoses [55]. As well as clinical appropriateness, adherence could become an issue for online course/programme delivery when COVID-19 restrictions are removed and people are no longer working from home [42]. To facilitate successful future virtual care, several factors should be considered. For example, appointments need to be scheduled in a pragmatic and logical way; with suitable detailed instructions, to manage expectations; with sufficient time for questions and with a cautious approach to self-care [49].

## 4. Discussion

This systematic review focused on the economic, social, health, and psychological implications of COVID-19 on cancer patients and the rapid expansion of telehealth services emerged as a key theme in response to the pandemic. Use of telehealth during the pandemic was particularly important for cancer patients and survivors who, due to their weakened immune system, were at greater risk of contracting infections and of developing more severe infections compared to the general population [3]. So, employing telehealth along with other public health guidelines and initiatives provided protection for this cohort. Despite the adoption of telehealth and other efforts, most healthcare systems paused essential inpatient and outpatient services during phase one of the pandemic. The result was missed or delayed cancer diagnosis and disrupted treatment leading to worsening health outcomes, quality of life, and in some cases mortality [29].

In the study, we briefly outlined the benefits and implications of telehealth with the purpose of mitigating the impact of the pandemic through reducing the risk of COVID-19 exposure for cancer patients and their families. The results showed that telehealth adoption seemed to “bolster” the delivery of healthcare services, providing a level of continuity of care during this highly uncertain time. However, it is evidenced that a “one size fits all” approach to telehealth is not appropriate to support the delivery of essential healthcare services for cancer patients. Albeit for its continued use, consideration of constraints is warranted to determine which patients and services may be successfully enabled by telehealth.

From an economic perspective, there were costs and benefits associated with the adoption of telehealth. Healthcare professionals’ emergency response to maintaining a line of communication with cancer patients and their families meant pragmatic decisions were made to ensure some level of service was provided at a time when patient protection against COVID-19 infection was a priority. Telehealth emerged as one key valuable tool available to healthcare providers, in an extremely limited toolkit of possible responses to the pandemic. During the early phase of the pandemic, for some service providers, telehealth adoption consisted of using existing available technologies such as telephone consultations and video calls [38]. This was the case for approximately 61% of the studies presented here. For the remainder, there are costs associated with setup and maintenance of telehealth technology [53], including data protection. Following the early phase, strategic investments were made in hardware and software solutions to support the delivery of telehealth beyond the initial emergency response [66].

The cost of telehealth to national health systems, healthcare professionals, patients, and their families differs across jurisdictions, owing to variations in financial reimbursement models for health services. While some articles included in this review briefly consider cost in terms of reducing the cost of travel, access, and quality of care for cancer patients, future research could establish the time and cost savings for cancer patients accessing telehealth services. While studies indicated that patients were prepared to pay for telehealth services in the future [51], to ensure sustainability going forward, as telehealth shifts to becoming more embedded as part of routine services, appropriate reimbursement mechanisms will be necessary. This will require research to move beyond traditional approaches to support this emerging model of care delivery to answer questions such as: who will pay, where, and for what services? The latter could also consider potential savings arising from fewer appointment “no-show” incidents.

Patients’ access to essential services via telehealth reduced the need for face-to-face interactions, when were stretched to capacity with COVID-19 patients or in anticipation of a surge of COVID-19 patients needing specialist care. The availability of telehealth services minimised the need for emergency admissions to acute services [65]. Reductions in non-attendance [42,49,52,54] also enabled the latter. Telehealth enabled the continuity of multidisciplinary decision-making for cancer patients with “virtual tumour boards” bringing together a range of medical disciplines for discussions on how to best care for a patient with cancer [33,35,56].

Telehealth provides beneficial spillovers for patients and their families. These include reduced travel time and efficiency gains with reduced waiting times from performing consultations from home with a loved one present. However, there are also costs and access barriers, including network issues and technology costs [46,63,64], which disproportionately impact vulnerable groups. The latter can include older adults [31,42,53,54,55], those with poorer literacy skills, and those from lower economic backgrounds. Additionally, for those with cancers at sites which could impact cognition and other functions, for example, brain tumours, telehealth may not be as beneficial.

From a social perspective, while some cancer patients and their families extolled the benefits of telehealth, the pandemic exacerbated existing challenges around equitable digital access and utilisation of healthcare services. This “digital divide” has disproportionately affected vulnerable patient populations, for example older groups [53]. In addition, the attitude of staff, patients, and their families to the adoption and diffusion of telehealth technology differs, owing to several factors. Perceived skill gaps and lack of available hardware, software, internet access, and technical support can impact successful telehealth adoption. If telehealth is retained in the delivery of routine oncology services, understanding the adoption barriers and facilitators is necessary to ensure that the lessons learned are not left behind. Researchers, healthcare decision makers, and policy makers should explore new opportunities to tackle these challenges. One option is to assess the potential of telehealth hubs as an approach to centralising telehealth services, thus overcoming issues relating to access and support.

Finally, from a psychological perspective, the switch to telehealth for some was associated with higher levels of cancer worry and feelings of isolation [49,62], whereby the lack of in-person access created mental health strain for patients and survivors who were forced to solely rely on telehealth communication. For some patients, this outweighs the comfort and support benefits of being at home. While this review only captured the first wave of COVID-19, if oncology telehealth were to continue, efforts to minimise these adverse psychological impacts are warranted. A user-centred approach is necessary to design and develop telehealth that supports quality patient-healthcare professional communication and relationship development in a virtual environment. New opportunities to co-create telehealth services should be identified to ensure the design of usable and accessible services for cancer patients. This is particularly important when a patient is newly referred to a service. Additional research should explore clinical workflows and opportunities to incorporate telehealth to complement existing services, so patients experience the benefits that were accrued during COVID-19 and healthcare professionals are formally and explicitly allocated the time necessary to deliver quality personalised telehealth services. This research should include key efficiency and effectiveness indicators of cancer services.

New telehealth solutions should be designed to adapt to the changing needs and preferences of patients and their families. For example, appointments need to be scheduled appropriately and with sufficient time to allow for questions and consideration of information with appropriate health care practitioners and sufficient support information [49]. Likewise, patients’ preferences for format matter must be considered as well, for example, telephone versus video calls. Moreover, the types of consultations for which telehealth is used need further exploration and consideration. For example, telehealth is suitable for some settings and services, such as counselling [42,47], replenishing medications, and for patients seeking information regarding their oncological treatments [51]. However, it is unsuitable for others, such as staging results and treatment decisions [55,57], which carry a significant psychological burden which could be exacerbated by telehealth. Emerging technologies such as artificial intelligence and machine learning techniques could be used to predict a patient’s changing healthcare needs, incorporating factors such as their living arrangement and ability to communicate in a private space, their mental and spiritual wellbeing, etc.

A more intelligent blended approach to designing services that incorporate both telehealth and traditional face-to-face consultations may be needed to ensure quality of care and positive patient outcomes. These findings correspond with existing research assertions that telehealth is not a panacea. While telehealth is not a universal solution to delivering cancer services during a pandemic or beyond, it facilitated continuity of care during a highly uncertain time for some.

## 5. Conclusions

This paper provides a systematic literature review to identify the economic, social, health, and psychological implications of COVID-19 on cancer patients during wave one, with the availability of telehealth services for cancer patients emerging as a key theme. This review is not without limitations. Most of the studies included in the review are single institutional studies, where telehealth was employed for a variety of reasons in various settings, with small sample sizes, so wide-scale adoption and satisfaction of telehealth cannot be determined. Furthermore, the methodologies employed yield potential biases. These include selection biases arising from convenience sampling [44] and observation bias owing to data collection methods [52]. However, some studies lacked outcome data [15], where collected self-reported data were relied upon [44] and validated instruments were lacking [43].

The time and scope of this review means only the first wave of the pandemic was assessed. While some initiatives have persisted and, in many cases, helped to mitigate the impact of the pandemic on adult cancer patients, it is likely other temporary measures have waned. This study reveals that there have been telehealth successes and failures; the lessons learned present a significant opportunity to reimagine the delivery of healthcare, leveraging telehealth as a complement rather than a substitute. This approach will enable healthcare systems globally to future-proof their operations by better preparing for unique unexpected disruptive events such as pandemics and natural disasters.

## Figures and Tables

**Figure 1 sensors-22-03598-f001:**
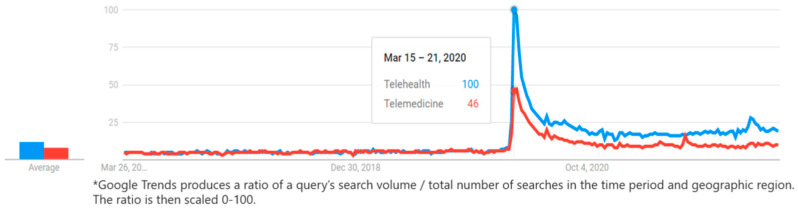
Google Trends search for “Telehealth” and “Telemedicine” (2017–2022).

**Figure 2 sensors-22-03598-f002:**
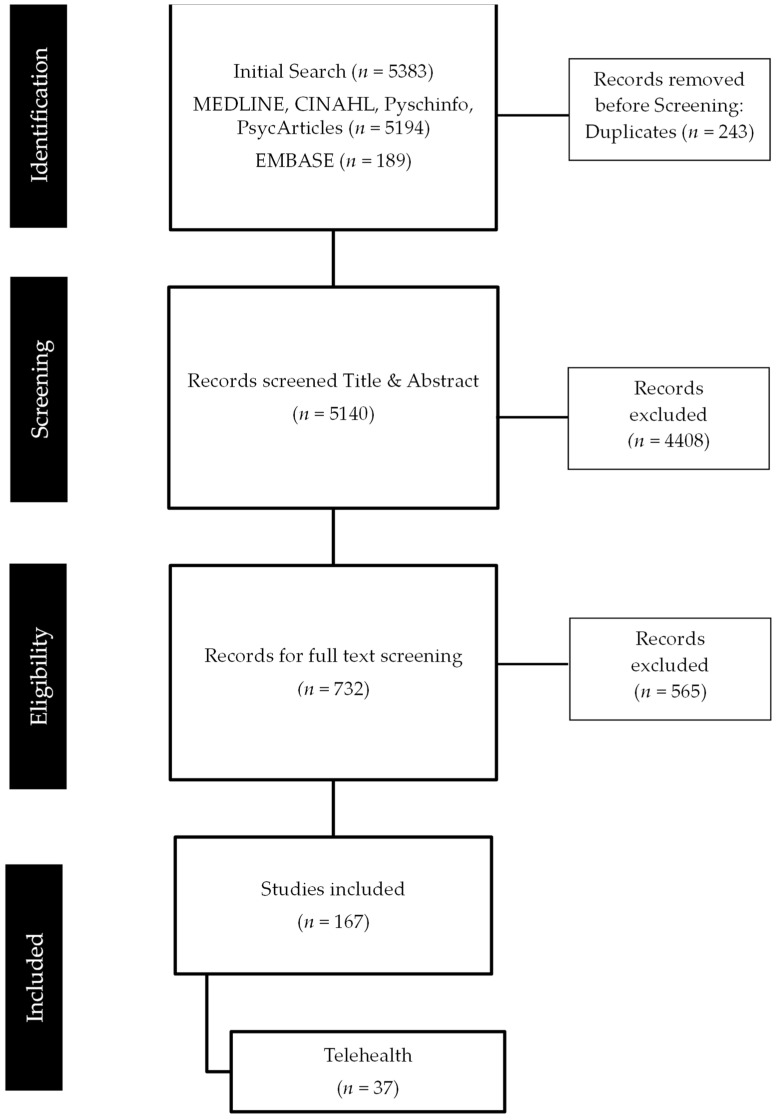
Search results.

**Table 1 sensors-22-03598-t001:** Inclusion criteria and search terms.

	Inclusion Criteria	Exclusion Criteria	Search Terms
Population	Adult population (>18 years old) Current cancer patients and survivors (2 years post-diagnosis)	Caregivers, nursing and medical staff, and paediatric cancer patients	“cancer” OR “oncology” OR “malignant” OR “tumour” OR “metastasis” OR “neoplasm”
Intervention	COVID-19 pandemic	-	“COVID-19” OR “coronavirus” OR “2019-ncov” OR “SARS-CoV-2” OR “cov-19” OR “severe acute respiratory syndrome coronavirus-2” OR “pandemic”
Outcome	economic, social, health, and psychological implications of COVID-19 on cancer patients/survivors	-	“financial toxicity” OR “out-of-pocket” OR “productivity” OR “absenteeism” OR “unemployment” OR “cost” OR “waiting time” OR “expenses” OR “financial stress” OR “inconvenience” OR “opportunity cost” OR “income” OR “wellbeing” OR “social isolation” OR “exclusion” OR “loneliness” OR “happiness” OR “life satisfaction” OR “fatigue” OR “insomnia” OR “psychological distress” OR “emotional distress” OR “anxiety” OR “depression” OR “post-traumatic stress disorder” OR “psychological” OR “quality of life” OR “health-related quality of life” OR “survival” OR “mortality” OR “disease progression” OR “diagnosis” OR “screening” OR “recurrence” OR “disease stage” OR “delay” OR “support” OR “surgery” OR “treatment” OR “target therapy” OR “radiotherapy” OR “chemotherapy” OR “immunotherapy” OR “hormone therapy” OR “survivorship programme” OR “follow-up-care”
Context	Hospital and community setting	-	
Studies	Full-text articles Patient perspective, Observational, Cross-sectional, Prospective, Longitudinal Retrospective	Letters to the editor, editorials, case studies, reports, protocols, commentaries, short communications, reviews, opinions, perspectives, and discussions	

**Table 2 sensors-22-03598-t002:** Overview of Review Papers (See Appendix C for further information).

Author Year Country	Aim	Telehealth Tool	Results
Akhtar et al. (2021) [38] India	Describe the hospital experience during the first 6 months of the COVID-19 pandemic.	Teleconsultations/virtual appointments for patients	Introducing teleconsultations decreased Outpatient Dept. workload.
Akuamoa-Boateng et al. (2020) [33] Germany	Compare hospital management in 2019 & 2020	Teleconsultations/virtual appointments for patients Video conferencing for staff	Hospital implemented telemedicine appointments for patients, a modified workflow and telemedical cancer board meetings via video call.
Alterio et al. (2020) [34] Italy	Report organisation strategies at a radiation oncology department, focusing on procedures and scheduling (i.e.: delays, interruptions)	Teleconsultations/virtual appointments for patients	Hospital transferred into an oncology hub & used telehealth for follow-up visit surveillance.
Araujo et al. (2020) [56] Latin America	Evaluate the impact of COVID-19 pandemic on patient volume in a cancer centre in an epidemic of the pandemic	Teleconsultations/virtual appointments for patients Video conferencing for staff	Offered telemedicine: virtual tumor boards, virtual consultations/appointments) in redesign of oncology care to replace face-to-face visits where possible. 45% reduction in medical appointments.
Atreya et al. (2020) [50] India	(1) Assess changes in the hospital-based practice of palliative care during the pandemic (2) Report patient/caregivers perception about the provision of palliative telehealth services (established 2014)	Teleconsultations/virtual appointments for patients	51% reduction in outpatient footfalls using telemedicine. 82% satisfied with advice given by palliative team. 64% felt comfortable using telehealth services. Telemedicine gave participants “support and connectedness”. 76% expressed their willingness to pay for telehealth service in the future.
Biswas et al. (2020) [51] India	(1) Assess expansion of service telemedicine in the palliative unit in the department of oncology (2) Assess patient satisfaction.	Teleconsultations/virtual appointments (telephone, texts and video) for patients	53.18% telephone calls and text messages. 26.75% required video consultations. Reasons for calling: symptom management; needed to restock opioid medications; for information regarding their oncological treatments. Patient satisfaction: 56 very satisfied; 152 satisfied; 59 partially satisfied; 47 unsatisfied; 42 patients believed that face-to-face consultations may be more useful for them
Brenes Sanchez et al. (2021) [39] Spain	Anlayse management breast cancer patients during the pandemic	Teleconsultations/virtual appointments for patients	Telemedicine faciliated evaluation of side effects and avoided unnecessary hospital visits. Patient perspective of quality of care of doctor and nurses: Technical skills, interpersonal skills, information administration and availability to patients. (>80%). Perspective of care management: Hospital staff interpersonal skills,, the exchange of information (77.6), waiting time (72%), hospital access and comfort (>70%).
Caravatta et al. (2020) [35] Italy	Report the experience and organisational planning of radiotherapy during pandemic	Telephone consultations. Telematics laboratory results Staff meetings on a telematic platform	Replacement of follow-up visits with telephone consultations. Laboratory and instrumental exams were viewed via telematics. Multidisciplinary Tumour Board meetings were held via telematics. Clinic re-opened for in-person visits after the second lockdown.
Clark et al. (2021) [48] England	Assess the national impact of COVID-19 on the prescribing of systemic anti-cancer treatment	Teleconsultations/virtual appointments for patients	Teleconsultations introduced as risk reducing measure at national level. Following an initial decline in registrations of new systemic anti-cancer treatments average monthly registrations exceeded pre-pandemic levels by June, 2020.
De Marinis et al. (2020) [36] Italy	Examine proactive management to minimise contagion among patients with lung cancer	Telephone consultations Email Telematics laboratory results	Adoption of telemedicine for follow-up visits (phone or email) Evaluation of CT scan imaging via telematics 100% of patients received triage phone call Follow-up visit cancellation was proposed to 50% of patients upon telematic consultation for radiology exam.
Earp et al. (2020) [40] USA	Examine the early effect of hospital and state mandated restrictions on an orthopedic surgery department	Teleconsultations/virtual appointments (telephone and video) for patients	Increased uptake of telemedicine (telephone encounter or video encounters) in Surgical Department: 0.3% to 81.2%.
Frey et al. (2020) [62] USA	(1) Evaluate the quality of life (QoL) of women with ovarian cancer during the pandemic (2) Evaluate the effects of the pandemic on cancer-related treatment.	Teleconsultations/virtual appointments for patients Online counselling Online networks	Online services included: telemedicine, counselling and survivor networks. 25% used telemedicine for gynecologic oncology care. Adoption of telemedicine was associated with higher levels of cancer worry.
Goenka et al. (2021) [31] USA	Review implementation of patient access to care & billing implications	Teleconsultations/virtual appointments (telephone and video) for patients	In-person visits decreased:100% to 21%. Telehealth appointments: 2-way audio-video (60%) or telephone (40%). Older patient less likely to have 2-way audio-video encounters. Inconsistent use of audio-video platform. Telehealth’s financial sustainability for all care questioned.
Kamposioras et al. (2020) [59] England	(1) Investigate the perceptions of service changes imposed by the COVID-19 pandemic. (2) Identify the determinant of anxiety in patients with colorectal cancer	Teleconsultations/virtual appointments (telephone and video) for patients	78% of participants had telephone consultation (83% met needs) & 6% had video consultation (80% acceptance rate). 40% had radiologic scan results discussed over the phone (96% met needs). Preferred consultation method: face-to-face 40% & 38% wanted a choice.
Kotsen et al. (2021) [42] USA	Examine the effect of rapid scaling to tobacco treatment telehealth for tobacco dependent cancer patients	Teleconsultations/virtual appointments for patients	100% of visits transferred to telehealth by March 2020. Increase in attendance: 75% for telehealth visits vs. 60.3% in-person visits. Telehealth visits had 2.30 times the odds of completion vs. in-person visit. Older aged patients had more challenges with telehealth setup. High patient acceptance with tobacco telehealth treatment. User-friendly telehealth platform is critical.
Kwek et al. (2021) [47] Singapore	Describe outpatient attendances and treatment caseloads during COVID-19 compared to pre COVID-19.	Teleconsultations/virtual appointments for patients and family members. Tele-counselling& psychosocial support. Medication delivery.	Increase in teleconsultation for surveillance follow-ups and outpatient consultations accounting for a 30.7% decrease in total face-to-face clinic consultations. Pharmacy department: tele-counselling & medication delivery. Telecommunication used for communication between families & patients in the palliative setting & with respect to advance care planning.
Lonergan et al. (2020) [15] USA	Analyse the change in video visit volume	Video consultations	Rapid expansion of telehealth (video consultations) from <20% to 72%. Video visits increased from 7–18% to 54–68%, between the pre- and post-COVID-19 periods. No disparity in uptake based on age, race/ethnicity, language or payer.
Lopez et al. (2021) [49] Canada	Describe adaptions to implement virtual cancer rehabilitation at the onset of the coronavirus disease 2019	Teleconsultations/virtual appointments (telephone and video) for patients	All in-person visits were rescheduled & converted to telephone visits (a secure 2-way videoconferencing telehealth platform). 221 referrals: decrease of 153 relative to the previous 3 months & increased over first 90 days; video appointments increased after the first 30 days. Increase or maintenance in the number of completed visits by appointment type vs. in-person care. Attendance rates ranged (80–93%) across visit types. Re: Access to care: increase access & attendance, patients receptive to telemedicine, increased programme capacity, communication barriers, challenges accessing a private space to discuss their health issues at home. Re: Meeting support needs: sense of reassurance and felt supported, helped cope with worries, some felt isolated by telemedicine. Re: Confidence with assessment and care plan: lack of in-person examination, relying on self-report/assessment of patients, worried about accuracy of describing symptoms, agreed video better than telephone visits, Both agreed preference for an initial in-person assessment.
Maganty, et al. (2020) [30] USA	Evaluate differences in patient populations being evaluated for cancer before and during the COVID-19 pandemic	Teleconsultations/virtual appointments for patients	Telehealth visits offered: Increase pre-COVID-19 to during-COVID-19 (1/585 versus 7/362) for screening and referrals. Cohorts were similar in terms of demographics and cancer sites.
Mahl et al. (2020) [63] Brazil	Evaluate delays in care for patients with head and neck cancer in post-treatment follow-up or palliative care during the COVID-19 pandemic	-	No report of telemedicine use. Cost of telemedicine acted as a barrier to care as they could not afford. teleconsultation technologies for palliative and follow-up services.
Merz et al. (2021) [43] Italy	Assess breast cancer survivors perceptions electronic medical record-assisted telephone follow-up	Electronic medical record-assisted telephone consulation/appointment.	80.3% satisfied with telephone follow-up vs. a standard follow-up visit. 89.8% satisfied with the duration of the phone call. 43.8% would like to have electronic medical record assisted telephone follow-up in the future. (median age was 62 years, 10% had a cancer previously, majority had early-tage breast cancer (68.3%)). No clinical indicators were associated with willingness to undergo future electronic medical record assisted telephone follow-up.
Mitra, et al. (2020) [64] India	Study the challenges faced by cancer patients in India during the COVID-19 pandemic	Teleconsultations/virtual appointments for patients	41.7% reported problems with slot availability for teleconsultation. 33% had network issues.
Narayanan et al. (2021) [44] USA	Report on the feasibility of conducting Integrative Oncology physician consultations via telehealth	Teleconsultations/virtual appointments for patients	842 patients in-person visits (April-October 2019); greater interest in discussing symptom management; & had worse self-reported ESAS symptom scores. 509 patients telehealth (consultations) (April-October 2020); wanted to discuss diet and nutrition exercise, herbs, and supplements. There was no significant difference PROMIS-10 score for mental health between the two cohorts in-person cohort reported worse physical health than the telehealth cohort.
Parikh, et al. (2020) [61] USA	Evaluate changes in resource use associated with the transition to telemedicine in a radiation oncology department	Teleconsultations/virtual appointments for patients	Telemedicine reduced provider costs $586 vs. with traditional workflow. Patients saved $170 per treatment course. Majority of consultations, follow up visits, and on-treatment visits were converted to telemedicine.
Patt et al. (2020a) [65] USA	Gain insights into the impact of COVID-19 on the US senior cancer population	Teleconsultations/virtual appointments for patients	Telehealth visits introduced, but limited scale owing to strain of COVID-19 & small oncology team… Telehealth visits did not offset the total reduction in in person Evaluation & Management services visits.
Patt, et al. (2020b) [54] USA	(1) Describe onboarding and utilization of telemedicine across a large statewide community oncology practice (2) Evaluate trends, barriers, and opportunities in care delivery during the coronavirus disease 2019 pandemic	Teleconsultations/virtual appointments for patients Virtual support groups (social workers provided) & tele-pharmacy	April–October 2020 telemedicine grew: 15% to 20% of new patient visits & 20% to 25% of established-patient visits. 96% of clinicians used telemedicine. 59% conducted new-patient visits with telemedicine. 64% reported telemedicine helped to expedite diagnosis & treatment more than seeing patients in person in the clinic. 55% of clinicians managed urgent issues by telemedicine. 80% believed that patients benefited from urgent assessment by telemedicine. 57% believed an emergency department visit or a hospital visit was avoided by telemedicine. 50% fewer no-shows versus face-to-face during COVID-19 Patient benefits: decreased exposure risk, decreased transportation. Barriers: Broadband access in rural areas & technical difficulties (older patients).
Patt et al (2021) [53] USA	Assess the implementation of multidisciplinary telemedicine in community oncology; providers and patients satisfaction; changes in clinic operations; opportunities and barriers	Teleconsultations/virtual appointments for patients	>50,000 telemedicine visits with patients by October: 15–20% of new patients and 20–25% of established patients. 76% satisfied with telehealth platform. Patients: desire to maintaining the telehealth option in the future; grateful and happy to have the option to visit their clinicians on a telemedicine platform; reduced distress. Challenges providers heard from patients: Older patient population technology hassle; 35% patients were frustrated with technology first-time use; Broadband access in rural areas; technical difficulties.
Rodler et al (2020) [57] Germany	Determine patients’ perceptions on adoption of telehealth as a response to the pandemic and its sustainability in the future	Teleconsultations/virtual appointments for patients Video conferencing	Adoption of telehealth & virtual multidisciplinary tumor boards via video conference. 62.6% of patients prefer to pursue in-person visits. Majority of patients were not inclined to continue using telehealth for staging results and treatment decisions. Patients on immunotherapy were less willing to continue with telemedicine in the future.
Romani et al. (2021) [32] Canada	Examine the effect of the COVID-19 pandemic on the operation of satellite radiation oncology facility and patient satisfaction	Teleconsultations/virtual appointments for patients	Successful adoption of telemedicine, increased use from 20.7% in 2019 to 100% in 2020. High patient satisfaction with telemedicine. A remote viewing system allowed radiation oncologists & physicians to remotely view alignment of computed tomography scans.
Sawka, et al. (2021) [37] Canada	Describe the management of small low risk papillary thyroid cancer during the COVID-19 pandemic	Telephone and video communciations.	6.8% patients had an in-person clinical or research visit during the pandemic (93.2% teleconsultations). 92.3% consented to telephone communication. 79.0% consented to videoconferencing communication. Advantages: reduced travel and waiting time & associated expenses for patients & caregivers; enables family members to attend; have “time and space” to make decisions in own environment. Challenges: communication issues with those who are hearing impaired, languages barriers, privacy considerations.
Shannon, et al. (2020) [45] USA	Determine how visit and genetic testing volume was impacted by new telephone genetic counselling and home testing.	New telephone genetic counselling and home testing.	Shifted to telephone genetic counseling. Maintained 99% of total visit capacity & decrease in no shows (9.5% to 7.3%) Fewer receiving telephone service consented to genetic testing compared to pre-COVID-19 period. 32% of the sample were not sent to laboratories. Reported obstacles: new sample required (missing sample, quality not sufficient, or mislabelled sample), non-enrolment in the online patient portal and technological difficulties.
Smrke, et al. (2020) [55] UK	Evaluate the impact of telemedicine on patients, clinicians, care delivery	Teleconsultations/virtual appointments for patients	75% of planned in-person appointments were converted to telemedicine. Face-to-face appointments remained for urgent patients. Clinicians found telemedicine efficient and indicated lack of physical examination did not often affect care provision; 83% indicated workload was the same as face-to-face; 83% indicated lack of video-based assessment was a barrier to care. Patients: High rate of patient satisfaction; Reasons for telemedicine preference: were reduced travel time, expenses, and convenience; 80% desired some telemedicine as part of their future care; 48% would not want to hear bad news using telemedicine & 20% would not want to hear any scan results on the telephone; Preference: Mostly telemedicine = 39%; Only telemedicine = 6%; Mostly face-to-face = 34%; Only face-to-face = 20%. Neither sex or education level impacted choice of consultation methods, though patients who preferred face to face only were slightly older (median age, 69 years vs. 58 years) than those who preferred at least some telemedicine.
Somani et al. (2020) [58] UK	Assess outpatient and telemedicine (phone and video) volume during the pandemic.	Teleconsultations/virtual appointments (telephone and video) for patients	2361 outpatient clinic slots were scheduled: 66.3% were virtual consultations; 20% face-to-face; 13.6% were cancelled. 57% of face-to-face consultations were related to flexible cystoscopy. 90% of cancellations were diagnostic flexible cystoscopy. Patient and clinician benefits. Longer effects on health outcomes is unknown.
Sonagli et al. (2021) [46] Brazil	Demonstrate how telemedicine was an efficient tool to maintain outpatient appointments for breast cancer patients follow up and surveillance	Teleconsultations/virtual appointments (video) for patients	49.4% decrease in outpatient appointments. 89% had appointment through telemedicine (video). Connection issues (10)(not influenced by age or socio-economic factors).
Wai et al. (2020) [41] USA	Explore the impact on surgical care of head & neck cancer patients		New patient referrals during COVID-19 decreased: 81 (45 via telemedicine) vs. Pre-COVID-19: 119. Time from referral to first visit (Pre-COVID-19: 22 days ± 50) v’s (COVID-19 period: 9.7 days ±8.7). No statistical difference between time from referral placement to evaluation.
Wu et al. (2020) [52] Taiwan	Assess smartphone enabled telehealth model for palliative care family conferences	Video conferencing	5 families rated video conferencing as good or very good (36%). 9 families were neutral (64%). 10 families were willing to use video conferencing again. 7 families would prefer to communicate with medical teams face-to-face. No statistically signficant socio-demographic differences were evident between those neutral or satisfied with telehealth service.
Zuliani et al. (2020) [60] Italy	Analyse COVID-19 related organisational changes.	Teleconsultations/virtual appointments (telephone) for patients	Telephone service: 90% of follow-up consultations & 40% of specialist visits. Acceptance of phone-based follow-ups and restaging visits perceived as ‘not very adequate’ (17%) or ‘not adequate at all’ (18%).

## Data Availability

The data presented in this study are available in Table 2 and [29].

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
