# Peer review of "Mitigating the Impact of the COVID-19 Pandemic on Adult Cancer Patients through Telehealth Adoption: A Systematic Review"

_sensors, 2022, doi:10.3390/s22093598_

Round 1

Reviewer 1 Report

Congratulations to the authors on a well-written review of telehealth's systematic use of health services.

The proposed work systematizes the knowledge in this field in a very comprehensive way. The work is all the more vital as most of the reports at the start of the COVID-19 pandemic were based on a case study. In my opinion, the work provides a solid basis for further in-depth analysis of the possibility of using telehealth, taking into account economic, social, health and psychological conditions.

Reviewer 2 Report

The paper deals with several strengths and weaknesses associated with telehealth in oncological patients. Although the  paper has unusual form it can be useful contribution to the field. I recommend for publication. 

Reviewer 3 Report

In this paper the authors analyze the impact of telehealth (in the context of COVID19 pandemic) on cancer patients. The manuscript is well written and covers important aspect with a social impact. Nevertheless I have minor comments:

  • in the results part I would suggesting adding more information about the demographic data of the patients. Was telehealth satisfaction related to the age? education? Were also particular type of patients more prone to telehealth (brain tumor vs others?).
  • what about caregivers satisfaction? 
  • in the discussion section I would suggest explaining more in details the possible future development of telehealth since COVID 19 pandemic may represent the springboard for these technologies 

Reviewer 4 Report

1- line 92: requires editing

2- line 105-106: requires editing

3- The conclusion should be simplified, and most of what is written in it should be transferred to the discussion.
